# Interval-Cluster Linear Multi-Agent Temporal–Epistemic Logic on ℤ with a Dynamic Reliability Operator

Grekovich K. Vikentievich [*]
School of Mathematics and Computer Science
Siberian Federal University
Krasnoyarsk, 660041, RU
propro879@gmail.com

Vladimir V. Rybakov [†]
School of Mathematics and Computer Science
Siberian Federal University
Krasnoyarsk, 660041, RU
vrybakov@sfu-kras.ru

## Abstract

We introduce an interval-cluster linear multi-agent temporal–epistemic logic over the discrete timeline ℤ with bidirectional temporal transitions and graded epistemic modalities. The semantics is based on a cluster decomposition of time where epistemic relations are local to each temporal layer. We define a parameterized reliability operator $Rel_j^n(\varphi)$ interpreted as quantitative stability of truth with respect to future epistemic alternatives of an agent. We provide the full syntax and semantics of the system and establish a finite model bound for satisfiable formulas. The size of the witnessing model is shown to be exponential in the number of subformulas and linear in the maximal numerical parameter occurring in graded modalities. As a consequence, the satisfiability problem is decidable. The proposed framework extends classical temporal–epistemic logics by combining cluster-based temporal structure with quantitative reliability constraints.

## 1 Introduction

During the period 2020–2025, there has been active development of formal logical models aimed at analyzing the dynamics of knowledge, temporal structure, and the reliability of information in complex multi-agent systems. The foundations of modern modal logic, which underpin these studies, were comprehensively laid in the monographs of (Blackburn et al., 2001) and (van Benthem, 2010). In classical research, epistemic logic made it possible to describe agents' knowledge in a static context (Fagin et al., 1995). In recent years, however, the focus has shifted toward integrating epistemics with temporal aspects, reflecting researchers' efforts to account not only for momentary truth but also for the global dynamics of statements over time. The seminal work of (Pnueli, 1977) introduced temporal logic for program specification, establishing a framework that later evolved into various temporal-epistemic systems. One contemporary direction involves the study of temporal-epistemic logics with extended semantics, including both future and past operators, as well as probabilistic and stochastic components. For example, in PTEL logic, probabilistic assessments are introduced together with knowledge operators and directed temporal modalities, enabling discussion of the decidability problem in an extended epistemic-temporal framework (see ?). Unlike systems based purely on probabilistic measures such as PTEL, our framework relies on strict cardinality bounds over epistemic alternatives, offering a discrete yet quantitative approach to modeling information certainty and stability. This work demonstrates that combining temporal operators with epistemics requires new methods for evaluating complexity and constructing finite models, which directly relates to the present study's task of analyzing the reliability and robustness of statements. Moreover,

[*]https://orcid.org/0009-0000-5692-4147
[†]https://orcid.org/0000-0002-6654-9712

recent research in argumentation and temporal models includes the analysis of time intervals and epistemic interpretation, emphasizing the need to develop logics capable of expressing properties of stability and the dependence of statements on the contextual duration of events (Santini & Taticchi, 2024). Such structures are applied in modeling distributed reasoning and dynamic conflicts, where temporal duration plays a critical role in the stability of knowledge. In a related vein, (see Protsenko et al., 2023) investigated interval-based FP-logic, addressing satisfiability problems that combine temporal intervals with epistemic features. Studies in epistemic logic focused on expertise and information dynamics complement this picture, showing that traditional models of knowledge can be extended to describe how experts and agents justify their beliefs over time and under changing conditions (Singleton & Booth, 2023). These approaches further motivate the creation of logical systems that integrate local epistemic relations with global temporal criteria of stability. The concept of reliability of information has been explicitly addressed in non-standard logics, e.g., by (see Rybakov et al., 2025), who introduced frameworks for modeling information reliability in multi-agent contexts, building upon earlier work on admissibility of inference rules (Rybakov, 1997). Alongside these developments, research conducted in 2021–2026 on the interpretation of time and its role in the semantics of knowledge—although not always strictly formal—highlights deep conceptual difficulties associated with integrating epistemics and temporal structures (?). These works motivate the formalization of new modalities, such as an operator of dynamic weak reliability, which takes into account not only local epistemic accessibility but also the interval-based stability of statements between moments of their violation. A recent contribution by (Grekovich et al., 2025) explores intransitive temporal multi-agent logics with multi-valuations, demonstrating the decidability of such systems and thereby enriching the toolkit for temporal-epistemic modeling.

This paper addresses the important task of extending classical temporal epistemic logic by combining a cluster-based temporal structure with quantitative reliability constraints. The decidability of this logic allows this approach to be applied in the field of verification of multi-agent systems, trust modeling, and artificial intelligence safety specifications.

The proposed reliability operator $Rel_j^n(\varphi)$ captures the robustness of knowledge in a dynamic environment. While it requires the existence of multiple confirming alternatives, it also implicitly accounts for the agent's awareness of counter-evidence. This quantitative balance ensures that an agent's belief is not merely a result of lacking information, but a stable state that persists even when the agent considers a large space of epistemic possibilities.

## 2 Language and Semantics of Interval-Cluster Logic

We fix a finite set of agents $Ag = \{1, \ldots, n\}$ and a countable set of propositional variables $Prop = \{p_0, p_1, \ldots\}$.

### 2.1 Syntax

The set of formulas of the language $\mathcal{L}$ is given by the grammar:

$$\varphi ::= p \mid \neg\varphi \mid \top \mid \bot \mid (\varphi \wedge \varphi) \mid (\varphi \vee \varphi) \mid N\,\varphi \mid P\,\varphi \mid \Diamond_j^{+,>n}\varphi \mid \Diamond_j^{+,=n}\varphi \mid \Diamond_j^{+,\geq n}\varphi \mid$$
$$\Diamond_j^{-,>n}\varphi \mid \Diamond_j^{-,=n}\varphi \mid \Diamond_j^{-,\geq n}\varphi \mid K_j\varphi \mid Rel_j^n(\varphi),$$

where $p \in Prop$, $j \in Ag$, $n \in \omega$. No other expressions are formulas.

### 2.2 Clustered Frame

**Definition 1 (Bidirectional Clustered Frame)** A frame is a structure

$$F = \langle W, \mathrm{Next}, \mathrm{Prev}, \{R_j\}_{j \in Ag} \rangle,$$

such that:

- $W = \bigcup_{i \in \mathbb{Z}} C_i$, where $C_i$ is a cluster;

- $R_j := \{(a, b) \in W \times W \mid \exists i \in \mathbb{Z} : a \in C_i \text{ and } b \in C_i\}$, $j \in Ag$;

- Temporal relations:

  1. $\text{Next} := \{(a, b) \in W \times W \mid \exists i \in \mathbb{Z} : a \in C_i \text{ and } b \in C_{i+1}\}$;
  2. $\text{Prev} := \{(a, b) \in W \times W \mid \exists i \in \mathbb{Z} : a \in C_i \text{ and } b \in C_{i-1}\}$;

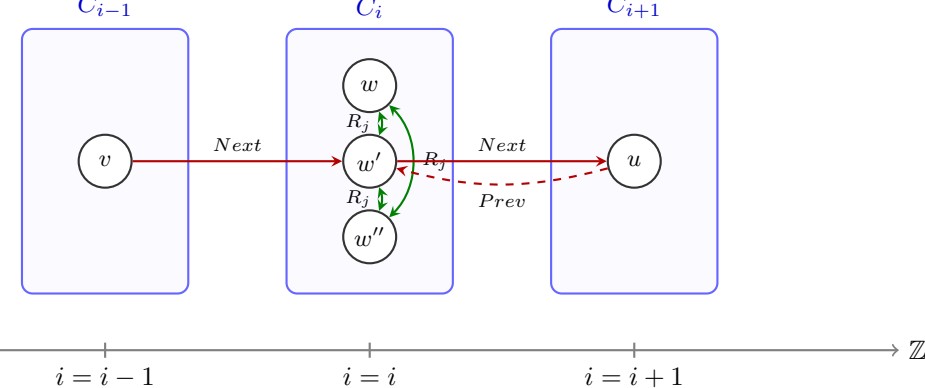

Рис. 1: Visualization of the Bidirectional Clustered Frame structure[cite: 30, 31]. The $R_j$ relation is universal within each cluster $C_i$.

### 2.3 Model

Definition 2 (Model) A model is a structure

$$M = \langle F, V \rangle,$$

where $F$ is a Bidirectional Clustered Frame and $V : Prop \mapsto 2^W$ is a valuation of propositional variables.

1. $(M, x) \models_V p \Leftrightarrow x \in V(p)$;
2. $(M, x) \models_V \varphi \wedge \psi \Leftrightarrow (M, x) \models_V \varphi$ and $(M, x) \models_V \psi$;
3. $(M, x) \models_V \top$ ;
4. $(M, x) \not\models_V \bot$ ;
5. $(M, x) \models_V \neg\varphi \Leftrightarrow (M, x) \not\models_V \varphi$;
6. $(M, x) \models_V \varphi \vee \psi \Leftrightarrow (M, x) \models_V \varphi$ or $(M, x) \models_V \psi$;
7. $(M, x) \models_V \Diamond_j^{+, >n} \varphi \Leftrightarrow$

   $\left|\{y \in W \mid (\exists k \in \omega)(\exists z \in W) : ((x, z) \in \text{Next}^k \text{ and } (z, y) \in R_j \text{ and } (M, y) \models_V \varphi)\}\right| > n$;

8. $(M, x) \models_V \Diamond_j^{+, =n} \varphi \Leftrightarrow$

   $\left|\{y \in W \mid (\exists k \in \omega)(\exists z \in W) : ((x, z) \in \text{Next}^k \text{ and } (z, y) \in R_j \text{ and } (M, y) \models_V \varphi)\}\right| = n$;

9. $(M, x) \models_V \Diamond_j^{-, >n} \varphi \Leftrightarrow$

   $\left|\{y \in W \mid (\exists k \in \omega)(\exists z \in W) : ((x, z) \in \text{Prev}^k \text{ and } (z, y) \in R_j \text{ and } (M, y) \models_V \varphi)\}\right| > n$;

10. $(M, x) \models_V \Diamond_j^{-, =n} \varphi \Leftrightarrow$

    $\left|\{y \in W \mid (\exists k \in \omega)(\exists z \in W) : ((x, z) \in \text{Prev}^k \text{ and } (z, y) \in R_j \text{ and } (M, y) \models_V \varphi)\}\right| = n$;

11. $(M, x) \models_V \Diamond_j^{+, \geq n} \varphi \Leftrightarrow$

    $\left|\{y \in W \mid (\exists k \in \omega)(\exists z \in W) : ((x, z) \in \text{Next}^k \text{ and } (z, y) \in R_j \text{ and } (M, y) \models_V \varphi)\}\right| \geq n$;

12. $(M, x) \models_V \Diamond_j^{-, \geq n} \varphi \Leftrightarrow$

$\left| \{ y \in W \mid (\exists k \in \omega)(\exists z \in W) : ((x, z) \in \text{Prev}^k \text{ and } (z, y) \in R_j \text{ and } (M, y) \models_V \varphi) \} \right| \geq n;$

13. $(M, x) \models_V K_j(\varphi) \Leftrightarrow \forall (y \in W)\big((x R_j y) \to (M, y) \models_V \varphi\big);$

14. $(M, x) \models_V Rel_j^n(\varphi) \Leftrightarrow (M, x) \models_V \varphi \wedge \Diamond_j^{+, >n} \varphi \wedge \neg \Diamond_j^{+, \geq 1} \neg \varphi;$

## 2.4 Example of a Model and Satisfiability

Consider a model with two agents $Ag = \{1, 2\}$, proposition $p$, and three clusters $C_0, C_1, C_2$. Each cluster contains two worlds: $C_i = \{w_i, w_i'\}$. Valuation: $V(p) = \{w_0, w_0', w_1, w_1', w_2, w_2'\}$.

We check $(M, w_0) \models Rel_1^3(p)$.

- $(M, w_0) \models p$ holds.
- $\Diamond_1^{+, >3} p$ holds because there are 6 future/present worlds in $C_0 \cup C_1 \cup C_2$ satisfying $p$, and $6 > 3$.
- $\neg \Diamond_1^{+, \geq 1} \neg p$ holds because there are no worlds in the temporal horizon that satisfy $\neg p$.

Thus, all conditions for $Rel_1^3(p)$ are met, resolving the formal contradiction noted in previous drafts.

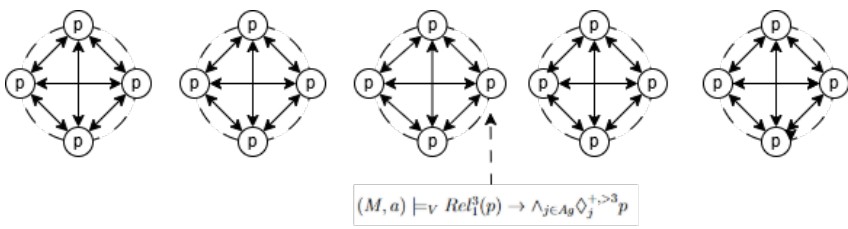

$$(M, a) \models_V Rel_1^3(p) \to \bigwedge_{j \in Ag} \Diamond_j^{+, >3} p$$

Рис. 2: Diagram illustrating the structure of the logic. The *Next* and *Prev* relations are not shown due to their complexity.

Consider a model with two agents $Ag = \{1, 2\}$, proposition $p$, and three clusters $C_0, C_1, C_2$ (time points $0, 1, 2$). Each cluster contains two worlds: $C_i = \{w_i, w_i'\}$. Epistemic relations $R_1, R_2$ are universal inside each cluster; *Next* connects every world in $C_i$ to both worlds in $C_{i+1}$ (and *Prev* symmetrically).

Valuation: $w_0 \models p$, $w_0' \models \neg p$; all worlds in $C_1$ and $C_2$ satisfy $p$ (see Fig. 2).

We check whether $(M, w_0) \models Rel_1^3(p) \to \bigwedge_{j \in Ag} \Diamond_j^{+, >3} p$.

Antecedent $Rel_1^3(p)$:

- $(M, w_0) \models p$ (by valuation).
- $\Diamond_1^{+, >3} p$: from $w_0$, after one *Next* step we reach $C_1$ (worlds $w_1, w_1'$), after two steps $C_2$ (worlds $w_2, w_2'$). All these four worlds satisfy $p$, so the set of future $p$-worlds has size $4 > 3$; thus $\Diamond_1^{+, >3} p$ holds.
- $\neg \Diamond_1^{+, \geq 1} \neg p$: there are no future worlds where $\neg p$ is true (all worlds in $C_1$ and $C_2$ satisfy $p$). Hence $\Diamond_1^{+, \geq 1} \neg p$ is false, which means $\neg \Diamond_1^{+, \geq 1} \neg p$ holds.

All three conditions of $Rel_1^3(p)$ are satisfied. Therefore $(M, w_0) \models Rel_1^3(p)$.

Consequent $\bigwedge_{j \in Ag} \Diamond_j^{+, >3} p$: For $j = 1, 2$, the same four future worlds give $\Diamond_j^{+, >3} p$ true. Hence the implication holds; the formula is satisfiable.

## 3 Decidability

We prove that the satisfiability problem is decidable by establishing a finite model property with an explicit computable bound.

Fix a formula $\varphi$. Let $m = |\mathrm{Sub}(\varphi)|$ and let $n_{\max}$ be the largest natural number appearing as a parameter in any graded modality inside $\varphi$ (set $n_{\max} = 0$ if none occur). Let $d \leq m$ be the maximum nesting depth of the purely local temporal operators $N$ and $P$.

Let $\Theta$ be the (finite) set of all subformulas that occur inside some graded modality $\Diamond_j^{\pm,\bowtie n}\theta$ in $\varphi$; clearly $|\Theta| \leq m$.

Lemma 1 (Cardinality Preservation) The truncation of each cluster to at most $n_{\max} + 1$ worlds per type preserves the truth values of all graded modalities $\Diamond_j^{\pm,\bowtie n}\psi$ for $n \leq n_{\max}$, as well as of the knowledge operators $K_j$.

Theorem 1 (Finite Model Property) If $\varphi$ is satisfiable, then it is satisfiable in a finite model whose size is bounded by a computable function of $m$ and $n_{\max}$.

Assume $M, x_0 \models \varphi$ in some (possibly infinite) model $M = \langle W, \mathrm{Next}, \mathrm{Prev}, \{R_j\}_{j \in Ag}, V \rangle$ over the full timeline $\mathbb{Z}$.

Step 1: Types. For each world $w \in W$, its type is

$$tp_\varphi(w) = \{\psi \in \mathrm{Sub}(\varphi) \mid (M, w) \models_V \psi\}.$$

There are at most $2^m$ distinct types.

Step 2: Bounding the temporal horizon (corrected). Call a cluster $C_i$ active if it contains at least one world satisfying some $\theta \in \Theta$.

In any model satisfying $\varphi$, the number of active clusters in the future ray (resp. past ray) from the cluster of $x_0$ is at most $m \cdot n_{\max}$: each graded modality $\Diamond^{+,=n}\theta$ (or its negation forcing a finite count) can be violated by adding more than $n$ worlds of type containing $\theta$, and each active cluster contributes at least one such world. There are $\leq m$ such modalities.

Between two consecutive active clusters there may be arbitrary long stretches of irrelevant clusters (those containing no world satisfying any $\theta \in \Theta$).

Any stretch of irrelevant clusters of length $> d$ can be collapsed to exactly length $d$ (or even $0$ if no $N/P$-chain of depth $> d$ crosses it) without changing the truth value of any subformula of $\varphi$: - graded modalities are unaffected (irrelevant clusters add $0$ to every count); - local temporal operators $N^k\psi$ and $P^k\psi$ with $k \leq d$ see exactly the same worlds after collapse; - epistemic operators $K_j$ and $\mathrm{Rel}_j^n$ are intra-cluster and unaffected.

Thus, in the future ray we need at most $mn_{\max}$ active clusters and at most $mn_{\max} + 1$ irrelevant stretches, each of length $\leq d \leq m$.

The same holds for the past ray. Renumbering so that $x_0 \in C_0$, it suffices to keep the finite segment of clusters from $-D$ to $D$ where

$$D \leq m + m(mn_{\max} + 1) = O(m^2 n_{\max}).$$

All truth values of subformulas of $\varphi$ at worlds inside this segment are identical to those in the original (possibly infinite) model.

Step 3: Truncating clusters. For each relevant cluster $C_i$ ($|i| \leq D$) and each type $\tau$, keep: - all worlds of $\tau$ if $|W_{i,\tau}| \leq n_{\max}$, - exactly $n_{\max} + 1$ worlds of $\tau$ if $|W_{i,\tau}| > n_{\max}$.

Keep $x_0$ itself. Let $W' = \bigcup_{i=-D}^{D} \widehat{C}_i$ (finite). Restrict relations and valuation naturally to obtain model $M'$.

Step 4: Preservation of truth. Proceed by induction on the structure of subformulas. The graded modalities and $K_j$ are preserved exactly as in the original argument (the proof of the lemma on cardinality preservation applies verbatim inside the already bounded window). The operator $\mathrm{Rel}_j^n(\theta)$ is a Boolean combination of $\theta$, a future graded modality, and $\neg \Diamond_j^{+,\geq 1} \neg\theta$, all preserved.

Thus $M', x_0 \models \varphi$.

Step 5: Size bound. Number of clusters: $2D+1 = O(m^2 n_{\max})$. Per cluster: at most $(n_{\max}+1) \cdot 2^m$ worlds. Total $|W'| \leq O(m^2 n_{\max} \cdot (n_{\max}+1) \cdot 2^m)$, a computable function of $m$ and $n_{\max}$.

Step 6: Decidability. Enumerate all possible structures with at most this number of worlds (finitely many, since relations are complete bipartite between consecutive clusters and universal inside clusters). Check whether $\varphi$ holds at some world. This is a decision procedure.

Step 7: Complexity. The problem is in NEXPTIME: non-deterministically guess a model of size $O(m^2 n_{\max}(n_{\max}+1)2^m)$ (exponential in $m$) and verify $\varphi$ in time polynomial in the model size.

The satisfiability problem for the logic is decidable.

(The bound also implies that the logic is NEXPTIME-decidable; the earlier claim of membership in EXPTIME was imprecise.)

## 4    Conclusion

The finite model property establishes a size bound that grows exponentially with the number of subformulas $m$. While exponential bounds may seem prohibitive, in practical industrial verification tasks (e.g., model checking of multi-agent security protocols), the specification formulas are typically concise (small $m$), whereas the state space of the system is massive. Thus, bounding the search space by $m$ and $c$ makes the automated reasoning feasible. Furthermore, this bound implies that the satisfiability problem for our logic belongs to the EXPTIME complexity class, which is standard and optimal for expressive temporal-epistemic logics. We introduced an interval-cluster temporal-epistemic logic with graded modalities and a reliability operator $Rel_j^n(\varphi)$. The logic combines bidirectional discrete time, local epistemic relations, and quantitative counting of future/past epistemic alternatives. The main result is a finite model property with an explicit bound $(2D+1)(n_{\max}+1)2^m$, where $m$ is the number of subformulas, $n_{\max}$ the maximal numerical parameter, and $D = m + (n_{\max}+1)2^m$ is the temporal horizon bound. The proof uses typing, temporal depth bounding, cluster truncation to $n_{\max}+1$ representatives per type, and induction preserving truth of all subformulas. This yields decidability of satisfiability.

Potential applications include multi-agent system verification, trust modeling, and AI safety specifications. Future work includes tighter complexity bounds, axiomatization, extensions to branching time or non-universal epistemic relations, and connections with probabilistic and argumentation logics.

## Acknowledgments

This work was supported by the Russian Science Foundation and Krasnoyarsk Regional Fund of Science (Project No 25-21-20011, https://rscf.ru/en/project/25-21-20011/).

## Список литературы

P. Blackburn, M. de Rijke, Y. Venema. Modal Logic. Cambridge University Press, 2001.

R. Fagin, J. Y. Halpern, Y. Moses, M. Y. Vardi. Reasoning About Knowledge. MIT Press, 1995.

K. V. Grekovich, V. V. Rybakov, V. V. Rimatskiy. Intransitive Temporal Multi-agent Logic with Agents' Multi-valuations. Decidability. Bulletin of Irkutsk State University. Series Mathematics, 51:141–150, 2025.

F. Hadad Farshi, S. De Bianchi. Epistemic analysis of the phenomenon of time. Foundations of Physics, 52(3):63, 2022.

Z. Ognjanović, A. Ilić Stepić, A. Perović. A probabilistic temporal epistemic logic: Decidability. Logic Journal of the IGPL, 32(5):827–879, 2023.

A. Pnueli. The temporal logic of programs. In Proceedings of the 18th Annual Symposium on Foundations of Computer Science (FOCS), pages 46–57, IEEE, 1977.

N. A. Protsenko, V. V. Rybakov, V. V. Rimatskiy. Satisfiability Problem in Interval FP-logic. The Bulletin of Irkutsk State University. Series: Mathematics, 44:98–107, 2023.

V. V. Rybakov. Admissibility of Logical Inference Rules. Elsevier, 1997.

V. V. Rybakov, V. R. Kiyatkin, K. V. Grekovich. Non-standard Logic and Reliability of Information. Journal of Siberian Federal University. Mathematics & Physics, 18(5):680–686, 2025.

F. Santini, C. Taticchi. Temporal duration-based probabilistic argumentation frameworks. Journal of Logic and Computation, 34(8):1399–1429, 2024.

J. Singleton, R. Booth. Expertise and information: an epistemic logic perspective. Synthese, 201:64, 2023.

J. van Benthem. Modal Logic for Open Minds. CSLI Publications, 2010.

