# OpenReview forum: "A Graded Multi-Agent Logic of Knowledge in Forward–Backward Time"
_mathai.club/MathAI/2026/Conference — 2026 Oral_

### Official Review · Reviewer_z91t · 2026-03-11
**Review of the Article: Interval-Cluster Linear Multi-Agent Temporal–Epistemic Logic**

**Rating:** 7
**Confidence:** 3

**Review:**

## Summary
The submitted work introduces a novel interval-cluster linear temporal-epistemic logic for multi-agent systems over discrete time with bidirectional temporal transitions and graded epistemic modalities. The key contribution is the reliability operator
which quantifies the stability of a proposition across future epistemic alternatives for agent.

 The authors provide:

    - Full syntax and semantics of the logic.
    - A finite model property with an explicit size bound.
    - A proof of decidability of the satisfiability problem.
    - A discussion of potential applications in multi-agent systems, trust modeling, and AI safety.

The logic generalizes classical temporal-epistemic frameworks by integrating clustered temporal structures with quantitative reliability constraints.

## Quality
    - Rigorous formal definitions of syntax, clustered frames, and semantics.
    - Clear finite model construction using type truncation and cluster bounding.
    - Provides a constructive decision procedure for satisfiability.
    - Well-referenced, situating the work within recent developments in temporal-epistemic logic.
    - Some notations and formulas are dense and may be difficult to parse.
    - The figure illustrating the model is missing, reducing clarity.
    - Proofs are often textual rather than structured as formal theorems or lemmas.

## Clarity
    - Introduction motivates the problem and situates the logic within contemporary research.
    - Stepwise explanation of finite model construction is clear and logically sequenced.
    - Syntax and semantics are explicitly defined.
    - LaTeX formulas in the text are sometimes broken or unconventional.
    - Notation for graded modalities is dense and requires careful reading.
    - Dense paragraphs could benefit from bullet points or shorter sentences.

## Originality
    - Combines temporal clustering with graded epistemic modalities in a novel way.
    - Introduces the dynamic reliability operator.
    - Finite model property with explicit bounds is novel for interval-cluster multi-agent logics.
    - Some techniques (finite model construction, type truncation) are extensions of established methods.
    - Novelty mostly lies in combining existing techniques rather than introducing entirely new mathematical machinery.

## Significance
    - Applicable to multi-agent system verification, AI safety, and trust modeling.
    - Provides a decidable framework, important for formal verification tasks.
    - Quantitative reliability enables robustness analysis in dynamic epistemic environments.
    - Practical significance may be limited until tools or automated reasoning support are developed.
    - Extensions to branching time or probabilistic models are mentioned but not implemented.

## Pros
    - Rigorous formalization of interval-cluster temporal-epistemic logic.
    - Introduction of the reliability operator for quantitative epistemic analysis.
    -  Decidability established with an explicit finite model bound:
    where m is the number of subformulas, is the maximum graded parameter.
    - Well-situated in current literature (2020--2025).
    - Illustrative example demonstrating semantics and satisfiability.

## Cons
    - Notation for graded modalities is dense and non-standard.
    - Figures illustrating cluster frames are missing.
    - Some proofs are textual rather than formally structured.
    - Readability could be improved with diagrams or bullet points in the introduction.
    - Practical implementations or computational experiments are not provided.

---

### Official Review · Reviewer_g5ub · 2026-03-12
**Interesting idea, but the central example is formally inconsistent and the proof is not sufficiently rigorous**

**Rating:** 4
**Confidence:** 3

**Review:**

**Summary.**
The paper introduces an interval-cluster linear multi-agent temporal--epistemic logic over the discrete timeline $\mathbb{Z}$. The framework combines bidirectional temporal transitions (Next/Prev operators), cluster-based semantics where epistemic accessibility relations are local to each temporal layer, graded epistemic modalities counting reachable worlds satisfying a formula, and a parameterized reliability operator $Rel^n_j(\varphi)$. The main theoretical result is a finite model property with an explicit bound of $(2m+1)(n_{\max}+1)\cdot 2^m$, where $m$ is the number of subformulas and $n_{\max}$ is the maximal numerical parameter in graded modalities. As a consequence, decidability of the satisfiability problem is claimed.

**Quality.**
The paper has the recognizable structure of a formal logic paper: syntax, semantics, an illustrative example, and a proof sketch for the finite model property. The overall proof strategy -- world typing, temporal depth bounding, cluster truncation, and truth preservation by induction -- is standard for this area and is reasonable at a high level. However, in the present manuscript several key issues prevent me from considering the contribution sufficiently reliable in its current form.

First, there is a direct formal contradiction in the key example. The reliability operator is defined as

$$
Rel^n_j(\varphi) := \varphi \wedge \Diamond^{+,>n}_j \varphi \wedge \Diamond^{+,>n}_j \neg \varphi.
$$

For this formula to hold, all three conjuncts must be true simultaneously. However, in Section 2.4 the authors explicitly state that $\Diamond^{+,>3}_1 \neg p$ is false in the given model, and immediately after that conclude that all three conditions of $Rel^3_1(p)$ are satisfied and hence $(M,w_0)\models Rel^3_1(p)$. This is a plain logical inconsistency: if one conjunct is false, then the conjunction is false. Since this is the central illustrative example for the main new operator of the paper, this error significantly weakens confidence in the formal correctness of the surrounding development.

Second, there is a conceptual issue with the semantics of the reliability operator itself. As defined, $Rel^n_j(\varphi)$ requires both more than $n$ future $j$-accessible worlds satisfying $\varphi$ and more than $n$ future $j$-accessible worlds satisfying $\neg \varphi$. In other words, the operator requires simultaneous abundance of confirming and disconfirming alternatives. This seems to model informational conflict, ambiguity, or instability rather than reliability in the usual sense. The paper does not provide a convincing semantic justification for why this should be interpreted as “reliability,” and the motivating discussion is too brief to remove this concern.

Third, the argument in Step 2 of the finite model proof is not sufficiently rigorous for a result of this strength. The claim that truth of a formula at a point depends only on worlds reachable within depth $d$ is standard for Boolean combinations of non-counting modalities. However, the graded modalities used in the paper are cardinality-sensitive: they depend not only on reachability, but on the number of reachable worlds satisfying a formula. Restricting attention to a finite temporal window may change these cardinalities in a way that affects truth values. The paper does not provide a lemma showing that the truncation preserves the relevant inequalities, or that counts above $n_{\max}$ can be safely collapsed for all formulas under consideration. Without such an argument, the finite model property is not fully established at the required level of rigor.

Fourth, the decidability claim is obtained by exhaustive enumeration of finite models up to the stated computable bound. This is acceptable in principle, but the paper gives no indication of the computational complexity of the satisfiability problem. Even a short discussion of possible upper or lower bounds would make the contribution more informative.

**Clarity.**
The paper is readable overall, but the presentation is not polished. There is an unresolved reference “Fig. ??” in Section 2.4, which indicates that the manuscript was submitted without a complete final pass. The bibliography header is written in Russian while the rest of the paper is in English, which is an avoidable inconsistency. In addition, Figure 1 is difficult to read and, according to its caption, omits some temporal relations due to complexity, which limits its explanatory value. These are not fatal issues by themselves, but they contribute to the impression that the manuscript is still not fully finalized.

**Originality.**
The paper combines several known ingredients -- cluster-based temporal semantics, bidirectional time over $\mathbb{Z}$, graded epistemic modalities, and a new reliability operator -- into a single formal system. This combination may be novel at the level of system design. However, the paper does not provide a sufficiently precise comparison with the closest related approaches, especially PTEL and other nearby temporal--epistemic systems, so it remains unclear what exact expressive or technical advantage is gained by the proposed extension.

**Significance.**
The topic is relevant to the conference, especially for tracks connected with temporal logic, epistemic reasoning, and trusted AI. If correct, finite model property and decidability would be meaningful results. However, in the current version the formal inconsistency in the main example, the unclear semantic motivation of the central operator, and the gap in the proof prevent me from viewing the paper as a sufficiently reliable formal contribution.

**Pros.**
- The topic fits the conference scope and is relevant to temporal--epistemic logic and formal reasoning in multi-agent systems.
- The paper has a recognizable formal structure and states nontrivial metatheoretic goals.
- The explicit finite-model size bound is a useful aspect of the presentation.
- The general direction of extending temporal--epistemic logic with graded operators is potentially interesting.

**Cons.**
- There is a direct contradiction between the definition of $Rel^n_j(\varphi)$ and the worked example in Section 2.4.
- The semantic interpretation of the reliability operator is not convincing: the current definition appears to model informational conflict rather than reliability.
- The proof of the finite model property does not adequately justify preservation of cardinality-sensitive graded modalities under truncation.
- The comparison with the closest related systems is too weak to clearly establish the precise novelty of the contribution.
- The manuscript contains visible signs of incomplete proofreading (e.g., “Fig. ??”, mixed-language formatting, weak figure readability).

**Recommendation.**
The paper contains an interesting formal idea, but in its current form it is not sufficiently convincing as a mathematical logic contribution. The central example is formally inconsistent, the semantics of the main new operator is not well justified, and the proof of the finite model property requires a more rigorous treatment. I therefore do not recommend acceptance in the present version.

---

### Official Review · Reviewer_1qLZ · 2026-03-13
**There are problems in the "A Graded Multi-Agent Logic of Knowledge in Forward–Backward Time" paper**

**Rating:** 5
**Confidence:** 3

**Review:**

This paper is devoted to solution of such important task as extending classical temporal-epistemic logics by combining cluster-based temporal structure with quantitative reliability
constraints. Decidability of this logic allows applying this approach in the fields of multi-agent system verification, trust modeling, and AI safety specifications.

This paper has the following disadvantages:
1) The bound of finite model property depends on m as exponential function. It is necessary to add such discussion in the paper as reasons why dependency on m does not lead to too big bound in the industrial tasks.
2) It is necessary to correct Russian "Рис" text to English "Рис" text.
3) It is necessary to correct Russian " Список литературы" text to English "Refrences" text.
4) It is necessary to add all authors in the following reference:

A. Perovi´c. A probabilistic temporal epistemic logic: Decidability. Logic Journal of the
IGPL, 32(5):827–879, 2023

5) It is necessary to correct 2010 year to 1997 year in the following reference:

V. V. Rybakov. Admissibility of Logical Inference Rules. Elsevier, 2010.

6) It is necessary to correct 59 article number to 63 article number in the following reference:

F. Hadad Farshi, S. DeBianchi. An epistemic analysis of time phenomenon. Foundations of
Physics, 52(3):59, 2022.

---

### Decision · Program_Chairs · 2026-03-14

**Decision:**

Accept (Oral)

**Comment:**

Dear Author(s),

On behalf of the Program Committee of the International Conference on Mathematics of Artificial Intelligence (MathAI 2026), we are pleased to inform you that your paper has been accepted for an oral presentation at MathAI 2026.

Your paper was evaluated through a rigorous two-stage review process involving both automated screening and expert review by members of the Program Committee. The reviewers recognized the quality and contribution of your work.

Presentation details:

- Format: Oral presentation (15–20 minutes + 5 minutes Q&A)
- Mode: You may present either in person (offline) at the conference venue in Sirius, Russia, or remotely via Zoom. Please indicate your preferred mode when confirming your participation.
- Conference dates: Marh 30 - April 3, 2026
- Website: https://mathai.club

Next steps:

1. Please confirm your participation and presentation mode by replying to this email mathai.club@yandex.ru no later than March 15, 2026 18:00 Moscow time.
2. If you plan to attend in person, the organizing committee will provide accommodation details separately.
3. Please prepare your final camera-ready manuscript according to the formatting guidelines available at https://mathai.club and upload it to OpenReview by March 15, 2026 18:00 Moscow time.

Should you have any questions regarding the program, logistics, or your presentation slot, please do not hesitate to contact us.

We look forward to your contribution to MathAI 2026.

With kind regards,

MathAI 2026 Program Committee
International Conference on Mathematics of Artificial Intelligence
https://mathai.club
OpenReview: https://openreview.net/group?id=mathai.club/MathAI/2026/Conference
Telegram: https://t.me/MathAI_club
Email: mathai.club@yandex.ru